# Which Strategies and Corresponding Competences Are Needed to Improve Supply Chain Resilience: A COVID-19 Based Review

**Jethro Kiers, Jaap Seinhorst** 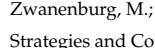**, Mirthe Zwanenburg and Klaas Stek ***

Faculty of Behavioural, Management and Social Sciences, University of Twente,
7522 NB Enschede, The Netherlands; j.b.kiers@student.utwente.nl (J.K.); j.seinhorst@student.utwente.nl (J.S.);
m.s.zwanenburg@student.utwente.nl (M.Z.)
***** Correspondence: klaas.stek@utwente.nl

**Abstract:** *Background:* During the COVID-19 pandemic, it became evident that supply chains were not as resilient as they should be. To cope with future disruptions or epidemic outbreaks, supply chain resilience should be improved based on lessons learnt from the impacts of COVID-19 to improve supply chain resilience and facilitate the corresponding competences and skills to implement strategic changes. *Method:* Applying the dynamic managerial capabilities theory lens, an integrative review is conducted focused on (1) the multiple impacts of COVID-19 on the supply chain resilience, (2) multiple strategies to enhance supply chain resilience, and (3) competences needed to implement the strategic changes successfully. *Result:* During COVID-19, supply chains could not handle supply and demand shocks, which led to a disrupted state of supply chains. To reduce the impacts of the subsequent disruptions, companies should implement specific strategic changes and increase complementary skills and competences levels. A proposed framework indicates which skills and competences need to be developed to implement a strategic change. *Conclusion:* Companies must shift their focus from (cost) efficiency to creating more resilient supply chains. Additionally, purchasing professionals should develop their skills and competences accordingly to cope with future disruptions.

**Keywords:** COVID-19; supply chain resilience; skills; competences; supply chain disruptions; impact; purchasing and supply chain management

## 1. Introduction

Since March 2020, after COVID-19 was declared a pandemic, lockdowns have been set in place to reduce the number of infections. Persons with symptoms were quarantined, professionals have worked in their home offices, and distance measures were enforced in all places. Many supply chains worldwide (86%) have been seriously impacted by the COVID-19 pandemic [1]. The direct consumer behaviours were hoarding of canned food, toilet paper and disinfection cleaning supplies due to the fear of stocked-out supplies [2]. This hoarding behaviour led to consumer stress and price and supply chain disruptions [2]. The problems grew when the shortage of materials was combined with a reduction in employees due to governmental lockdown actions to counter the number of infections. Suppliers could not meet delivery agreements, supply decreased due to closed factories, and customer demand became highly unpredictable [3]. It became evident that most global supply chains' resilience was under the desired levels [2]. The resilience of supply chains needs to be improved with the help of the lessons learned from the COVID-19 pandemic to decrease the impacts of future pandemics. Some essential questions are: which supply chain disruption occurred during COVID-19, which changes are needed to overcome the disruptions at the next pandemic, and which skills are required to implement these changes?

Whereas much research is available on themes such as (1) the impacts of the COVID-19 pandemic, (2) resilience strategies, (3) the role of technology in implementing resilience strategies, and (4) the COVID-19 pandemic and supply chain sustainability [4–6], no research is available on (A) which strategies can deal most effectively with which impacts and (B) the challenges and requirements associated with implementing resilience strategies [4]. This paper bridges the missing links between the impacts, strategies, and required skills to implement the strategies. Additionally, this paper elaborates on multiple impacts and strategies to improve supply chain resilience instead of just one impact or strategy.

This study focuses on the required inputs of professionals, and consequently human competences, consisting of a balanced mix of knowledge, skills, and attitudes, which are essential sources of organisational effectiveness [7–11]. This study depends on the elaborate definition of Delamare-Le Deist and Winterton [7] (p. 39): "The competences required of an occupation include both conceptual (cognitive, knowledge, and understanding) and operational (functional, psycho-motoric, and applied skill) competences. The competences more associated with individual effectiveness are also both conceptual (meta-competence, including learning to learn) and operational (social competence, including behaviours and attitudes)".

This article finds an answer to the research question: *Which strategies and corresponding competences are needed to improve supply chain resilience?*

The following research questions are formulated to answer the central question:

- What have been the impacts of COVID-19 on supply chains?
- Which strategic changes can be made to supply chain management to improve the resilience of supply chains in the post-COVID-19 time?
- What competences and skills are needed to implement strategic changes strategies to improve the resilience of supply chains in the post-COVID-19 period?

This paper contributes to the current literature by linking the impacts of COVID-19 with corresponding strategic changes to improve supply chain resilience in combination with the competences and skills needed to implement the proposed strategic changes. This study uses the dynamic managerial capabilities theory lens [12]. Dynamic managerial capabilities are "the capabilities with which managers build, integrate, and reconfigure organisational resources and competencies" [12] (p. 1012). The dynamic managerial capabilities "are rooted in three underlying factors: managerial human capital, managerial social capital, and managerial cognition" [12] (p. 1013). These factors influence the manager's perspective on strategic choices [12]. Therefore, this theory is applied to determine how purchasing and supply chain management (PSM) professionals develop their capabilities to increase their judgement towards strategies and supply chain resilience.

The paper's remainder is structured in the following way. First, Section 2 explains the methodology used during this integrative review. Secondly, Section 3 provides a theoretical background on supply chain resilience in general. Section 4 provides the integrative review. In Section 5, the findings of the integrative review are discussed, while in Section 6 the conclusion is provided. Finally, Section 7 states the managerial implications.

## 2. Methodology

For this article, an integrative review was conducted. Snyder [13] states that an integrative review addresses mature or new and emerging topics. It should result in an extension of knowledge and theoretical frameworks [13]. This article addresses the emerging topic of COVID-19 in combination with supply chain resilience strategies and the corresponding competences and skills. While there is no strict guideline for conducting an integrative review, the basic steps of a literature review should be conducted [13,14]. The steps that were conducted were as follows: (1) review design; (2) literature search; (3) literature analysis; (4) writing of the review [13,14]. For this methodology section, steps 1–3 are described. The remainder of this article relates to step 4.

This article's purpose, scope, and research questions are already stated in Section 1. The paper intends to link the impacts of COVID-19 with strategic changes to improve supply

chain resilience in combination with the competences and skills needed to implement the proposed strategic changes. The main research question is: *Which strategies and corresponding competences are needed to improve supply chain resilience?*

The electronic databases Scopus and Business Source Elite were used for this review. The search string that was used in Scopus was TITLE-ABS-KEY ("COVID-19" OR "pandemic" OR "corona" OR "coronavirus") AND ("supply chain disruptions" OR "supply chain improvements") AND ("risk management" OR "resilience" OR "disruptions" OR "vulnerability"). The search string that was used in Business Source Elite was KW (COVID-19 or pandemic or corona or coronavirus) AND KW (supply chain disruptions or supply chain improvements) AND KW (risk management or resilience or disruptions). With Scopus 189 articles were found, and with Business Source Elite 18 articles were found.

Only English written articles were included in identifying the relevant literature. This reduced the number of relevant articles from Business Source Elite to 16. The relevant articles found via Scopus remained at 189. Furthermore, sector-specific research (food or healthcare industry) was excluded from the search. This reduced the relevant articles found in Scopus to 148 and for Business Source Elite to 13. Then, duplicates were removed. The relevant articles that remained to be reviewed numbered 152.

After creating the review design, the literature was searched accordingly. First, the title, abstract, and conclusion were read to assess the relevance of an article. Based on this assessment, articles were added to the database or were left out of the research. After a solid base was created, the relevant literature was thoroughly assessed to determine whether the literature would still be relevant for this article. Additionally, the reference lists of the found literature were examined to identify more relevant literature.

The literature selected for this article addressing COVID-19 supply chain issues was analysed based on the unique contributions of the article. Other papers were not assessed based on unique contributions, since these articles did not contribute to the main focus of this research as depicted in the research questions. Table A1 shows analyses based on the unique contributions of each article.

## 3. Theoretical Background

This section describes the theoretical background needed to answer the research questions. Firstly, the theory of resilience and how to measure it is described. Secondly, we explain how supply chains have been changing in terms of resilience.

Many different definitions of resilience can be found between resilience in general and supply chains in the literature. Supply chain resilience "measures the ability to prepare for and provide essential functions during a disruption, and then to recover from and adapt post-disruption into a form that is better suited to the new 'present'" [15] (p. 223), stating that "a resilient system can withstand a disruption" [16] (p. 78). A systematic literature review on the definition of resilience showed that "many of the (definitions in literature) focus on the capability of a system to 'absorb' and 'adapt' to disruptive events, and 'recovery' is considered as the critical part of resilience" [17] (p. 49). Therefore, the most comprehensive definition will be used in this study: "Resilience is the ability of a system to detect, adapt, and react to disturbances to restore its original structure and functions" [18] (p. 1).

Given the definition of resilience above, metrics for resilience should incorporate the ability to detect, adapt, and react to disturbances. The systematic literature review performed by Hosseini and Barker [17] also focused on different methods to measure resilience. They found many different qualitative and quantitative ways to measure resilience. Most of the ways to measure resilience mentioned are metrics that use a combination of changes in performance after the disruption has taken place and the time taken to climb back to the original performance level [17]. Therefore, the essential factor for determining resilience is the time needed for a system to fully recover from the disturbance.

Contrary to the definition, time to recovery plays an essential role in determining resilience quantitatively. The time needed to recover depends on many different factors, for

example response time and quality of response. It also depends on the type and size of the disturbance. For instance, a significant fire in a factory of a strategic supplier is harder to recover from than the bankruptcy of a single non-critical supplier. A measurement method that can assess how resilience is changed from a PSM perspective in a more general context is Kraljic's matrix [19]. This matrix is a tool used to classify suppliers by their risk and profit impact. The classification helps the PSM department deal with the supplier properly.

The matrix has four quadrants: the non-critical items, which have low supply risk and low-profit impact and are on the lower-left quadrant; the lower-right quadrant contains the items with a low profit impact but high supply risk—the bottleneck items; the upper-left quadrant represents the items with a high profit impact but a low supply risk—the leverage items; lastly, the upper-right items are the strategic items with a high supply risk and high profit impact. Kraljic [19] describes dealing with suppliers of those items in the different quadrants. The Kraljic matrix [19] will be used to show how resilience is changed. Supply resilience was reduced due to suddenly increased supply risk item 'transfer' from the non-critical to the bottleneck quadrant and from the leverage to the strategic quadrant.

The most critical causes resulting in the current situation will be researched in the following paragraphs. Since the emerging interest in Industry 4.0, supply chain management has been an evolving topic. The motto is that change is the only constant [20]. This implies that supply chain management is constantly changing and evolving. Since supply chains have become increasingly complex, managing supply chain risk has become progressively vital [20]. Many senior supply chain executives classify supply chain risk as one of the biggest challenges currently, despite the majority having risk monitoring processes [21].

Two factors have been defined as critical for supply chain resilience [20]. The first factor is supply chain complexity. Business environments are becoming more turbulent due to the increases in globalisation and climate change globally. This has led to an increase in disruptions in supply chains. Organisations have more outsourcing opportunities, and the dependence on suppliers has increased. For example, non-critical and leverage products can also be sourced more cost-efficiently from Asian countries. Therefore, supply chains have become geographically longer, and supply risk has increased due to possible disruptions in the logistical chain.

The second factor is the realisation that traditional risk management is not contributing to more resilient enterprises. The commonly used enterprise risk management (ERM) method appears to be too simplistic to analyse complex supply chain risks. Most organisations, therefore, require new strategies to deal with increased turbulence in supply chains [20]. Moreover, supply chain management is generally optimised based on efficiency, not resilience [15].

## 4. Literature Review

### 4.1. The Impacts of COVID-19 on Supply Chains

COVID-19 impacted global supply chains and highlighted the need for building more resilient supply chains [22]. These impacts are (1) supply shocks, (2) demand shocks, (3) the bullwhip effect, and (4) transportation requirements and costs [22].

- **Supply shocks**

Social distancing and lockdown policies affected the movement of people and business operations [22,23]. This led to a sudden change in the supply of products [23]. Especially production facilities in Asia were closed during COVID-19 because of employees being in lockdown [22]. Due to outsourcing, many companies rely on Asian production facilities. As a result, manufacturers and retailers worldwide who relied on these facilities did not have access to enough raw materials and products for their businesses, or the lead times became too long [22,24]. Hence, production and manufacturing capabilities decreased, so supply shocks occurred [24].

- **Demand shocks**

Demand shocks are sudden changes in demand. Sectors such as culture, sports, and leisure experienced decreased demand since events and international travelling were prohibited or discouraged [25]. Additionally, consumers tried to reduce the risk of exposure to the virus and decrease demand for products and services that involve close contact with others [23]. In contrast, sectors such as IT and medical equipment manufacturing experienced increased demand because people needed to work from home and medical equipment is needed to prevent and cure people of COVID-19 [25]. A shift occurred from products considered as non-critical or leveraged to bottleneck and strategic products, especially for medical supplies, during COVID-19.

- **Bullwhip effect**

Consumers demand an increasing significant volume of scarce resources when recognising supply shortages. This leads to increasing swing inventories due to shifts in customer demand, which leads to more unpredictable demand patterns during COVID-19 [22]. One of the main reasons for the bullwhip effect is the lack of transparency, visibility, integration, and coordination [22,24]. Consumers are unaware of a supply shortage since information and data from partners are non-transparent [24]. Therefore, the anticipated supply shortages are reflected in increased volume orders. This bullwhip effect leads to unpredictable demand patterns during COVID-19 and higher prices [22,24].

- **Transportation and requirement costs**

Due to border and quarantine regulations, it was challenging to transport goods to other countries [22]. Moreover, cancelled passenger flights led to restrictions, since these aircraft also transport raw materials and products. This led to higher transportation requirements, costs, and delays [22,24].

Another general problem is the lack of disruption plans associated with lower inventory levels, single suppliers, minor diversification, and underestimating the possibility of disruptions [24]. As discussed, companies performed risk analyses to analyse risks, but these analyses did not deal with disruptions in supply chains and did not contribute to more resilient supply chains. Supply chain managers aim to have minimal inventories and single suppliers. They generally optimise based on (cost) efficiency and not resilience.

The result of all five impacts is reduced return, profit, and income [24]. Due to the decrease in supply and increase in demand, smaller companies especially did not have the resources to adapt to this new situation. Therefore, they had to close down and partly dismiss personnel. Additionally, the profits and returns from companies decreased since they could not produce as much as they wanted to due to shortages in raw materials.

*4.2. Possible Strategic Changes to Improve the Resilience of Supply Chains*

As of mid-October 2021, 47.8% of the global population has had at least one vaccination of a COVID-19 vaccine [26], and companies are focused on establishing more resilient supply chains [22]. Companies can search for measures to build a more resilient supply chain to answer the research question. The measures companies describe taking or planning to increase their resilience and risk management are listed below [22,24].

- **Diversification and Dual Sourcing**

Senior management personnel have forced supply chain managers to find other sources not dependent on China [22]. Investors and governments warned companies not to over-rely on any source, thereby diversifying the risks [22]. Additionally, supply shocks can be reduced by working closely with the current suppliers and diversifying the suppliers [24]. Companies are urged to find multiple suppliers instead of relying on a few suppliers. Having multiple suppliers for a product reduces the supply risk and moves the items that those suppliers supply (for a Kraljic perspective, see Kraljic's matrix [19]) from the bottleneck (strategic) quadrant to the non-critical (leverage) quadrant.

- **Vertical Integration of Supply Chains**

    "Vertical integration is a strategy that allows a company to streamline its operations by taking direct ownership of various stages of its production process rather than relying on external contractors or suppliers" [27] (p. 1). When companies are integrating vertically, they are acquiring the companies that are producing or supplying the goods they used to procure. Zhu and Chou [22] (p. 5) state that "bigger companies will also integrate vertically throughout the value chain to seize autonomy over costs, quality, and supplies and inputs". A company can decide how much risk they are willing to take or increase their resilience.

- **Decentralisation of Manufacturing Capacity**

    A trend was found among companies to decentralise manufacturing capacity. This means to localise production, deploy automation, and manufacture smaller batches to reduce costs, since these are now cheaper options [22]. Decentralisation also reduces risks and is a fast way to increase resilience.

- **Emphasis on Supply Chain Visibility**

    "Companies should be aware of their entire supply chain network and the interplay between their supply chain players to effectively plan for and mitigate risks in the event of such a disruption such as COVID-19" [22] (p. 5). "The visibility of the inventory in the entire SC allows organisations to integrate their efforts, database, and decision process for the necessary flexibility and agility to react to rapid changes, speed up processes, and reduce costs" [24] (p. 14). Supply chain visibility can be achieved by mapping the entire supply chain and data sharing process (among others inventory levels). This will help companies predict and prepare for disruptions from the supply side and the suppliers to predict and prepare for disruptions from the demand side. Additionally, the bullwhip effect that can occur after or during a disruption can be reduced by increasing visibility, since one of the main reasons for the bullwhip effect is a lack of visibility [22,24].

- **Localising Supply Chains**

    To reduce the reliance on China, "it will be a wise decision to localise some of the parts of the supply chain, if not all" [22] (p. 5). It is required to have nearby local or regional sources to stabilise supply chains [24], and an essential factor in handling a pandemic is lead time [28]. Localising supply chains will decrease the lead time since the absolute distances that need to be travelled will be decreased. Furthermore, reliance on local and regional sources leads to higher responsiveness to disruptions [24]. Localising supply chains fits the trend of consumers being more mindful of the environmental and ethical aspects of supply chains and that reducing the number of tiers in the supply chain can accelerate business processes [22].

- **Merging of B2B and B2C and Flexibility in Supply Chains**

    Companies increasingly need to develop flexible supply chains to respond to changing demands. During the COVID-19 pandemic, it has become evident how some companies were able to adapt quickly by changing their products or services to the market's needs [22]. By enabling this kind of flexibility and being able to supply to both the B2B (business-to-business) and B2C (business-to-customer) markets, companies might "be able to thrive in uncertain and volatile environments and outperform their capabilities while also discovering new sales and growth opportunities" [22] (p. 6).

- **Investing in Online Distribution Channels**

    The trend towards online shopping of consumers increased during COVID-19 and is still intensifying. Companies should "explore opportunities to reach out to customers virtually" [22] (p. 6). Research (not focused on COVID-19) has shown that e-commerce initiatives increase the average firm's value in a short period [29]. Therefore, investing in online distribution channels would be valuable for firms even without pandemic issues.

- **Digital Transformation of Supply Networks**

"Industry 4.0 technologies, SC 4.0, and augmented reality can expedite SC digital transformation" [24] (p. 14). Additionally, "digitisation enables risk management and business continuity as part of the entire business strategy" [24] (p. 14). Companies want to transform their traditional supply chains towards digital supply networks (DSNs) [22,30]. In these DSNs, there should be a "free flow of information and end-to-end visibility, corporation, dexterity, and optimisation of the supply chain" [22] (p. 6). Digitisation helps build resilient global supply chains [24]. Digitisation is a solution towards a more resilient supply chain, since visibility in a supply chain is vital, which can only be achieved by digitisation [22].

- **Government Policies and Assistance**

Governments outside China encourage companies to diversify their supply chains by giving tax advantages [22]. This is an external measure; a company has a meagre influence on this measure. However, since the pandemic caused an economic crisis, the government can play an essential part in the recovery. Governments can also use this measure to attract more activities to their country, resulting in extra tax income in the long term.

A qualitative study found six key measures "to overcome or at least mitigate the negative effects of the challenges (of COVID-19)" [31] (p. 65) in healthcare procurement systems. They categorised the measures into three categories. The measures taken are listed below.

- **Supply-side measures**

*Increase resilience through stockpiles*—An increased volume of safety stocks means less risk of a stock-out. By increasing safety stocks, inventory is held closer to the market. If a disturbance occurs at the supplier, a buffer and more time is available to overcome the disturbance [31].

*Increase domestic production capacity*—National production in the EU is more expensive than production in countries such China. However, producing close to the market decreases risks and increases resilience significantly [31].

- **Capability measures**

*Setup integrated information systems for data sharing*—Companies can react to disturbances earlier by sharing data. Suppliers from multiple tiers can react earlier to disturbances such as demand shocks. Data sharing improves visibility, which is (as mentioned earlier) an essential factor in improving resilience [31].

*Shift towards category management*—"Aim to increase market intelligence and category strategies through skills and competences" [31] (p. 65). Category management and strategic PSM help increase knowledge of the geographical spread and the impacts of visibility on the supply chain. Additionally, category management is essential for implementing dual sourcing [31].

- **Coordination measures**

*Setup an organisation for increased central procurement power*—In the case of healthcare, the aim is to increase central procurement power to provide more negotiation power and create a preferred customer status at the supplier. This makes sense when healthcare is partly public. In other sectors, this could be implemented by centralising the PSM department instead of having multiple departments independently purchase certain materials (if not already implemented) or by collaborating with companies that purchase the same materials [31].

*Establish crisis procurement protocols*—Learning from mistakes made during the last crisis and assuring that knowledge from the lessons learnt is used when a new crisis happens. The best way to plan for a future risk management system is to understand the risk drivers, priorities, and solutions [24]. "Organisations should study the potential

risks in other countries and regions to specify the actions and plans that will appropriately protect against these risks" [24] (p. 14).

An important note is that "the measures on their own cannot overcome or diminish the challenges encountered during the pandemic" [31]; they are interdependent. Since this study focuses solely on the healthcare procurement systems, we assess whether the measures are also applicable to other sectors. All measures can be applied in other sectors. However, setting up an organisation for increased central procurement power is less likely to succeed in the competitive private sector than in the less competitive public sector.

The abovementioned measures all indicate changes in supply chain management. Contrary to the pre-COVID trend that supply chains optimise based on (cost) efficiency, the trend should change towards giving resilience a higher weight in the optimisation strategies of supply chains.

Supply chains must become flexible, traceable, transparent, persistent, responsive, globally independent, and equitable to become immune [32]. The main difference between the abovementioned measures to become more resilient and the measures to become immune is intensive cooperation between organisations in times of crisis, which is more or less an extension of supply chain visibility.

In the third section, the multiple impacts of the COVID-19 pandemic are indicated. Since this study looks for lessons learned from the pandemic, an overview of which strategic changes can prevent or reduce the impacts of a pandemic such as that resulting from COVID-19 is given in Table 1.

**Table 1.** Connections and explanation of impacts of COVID-19 and corresponding strategic changes.

| Impact of COVID-19 Pandemic | Main Strategic Change to Prevent or Reduce Impact | Reason for Change to Prevent or Reduce Impact |
| --- | --- | --- |
| Supply shocks | Diversification/Dual sourcing Localising supply chains | The supply shocks were mainly caused by relying too much on a few suppliers, mainly from Asia. By diversifying and localising suppliers, supply shocks can be prevented or at least reduced. |
| Demand shocks Bullwhip effect | Emphasis on supply chain visibility Increase resilience through stockpiles | By increasing visibility within supply chains, customers can see whether enough stock is available in the chain. Therefore, less panic buying will reduce the magnitude of demand shocks and the bullwhip effect. Additionally, increasing safety stocks reduces the magnitude of the demand shocks and the bullwhip effect. |
| Transportation requirements and costs | Localising supply chains | The impact was caused by having many suppliers far from the market. By localising the supply chain, transportation costs will decrease, and the transport has to cross fewer borders, decreasing the requirements and risks. |

There are four levels of preparedness for a health crisis, with the third being resilience and the fourth being immunity [33]. Therefore, a resilient supply chain might not be the end goal for an organisation. After an organisation is resilient, it should focus on becoming immune. To achieve immunity from shortages in personal protective equipment (PPE), organisations must work together with other organisations to become more transparent, increase their visibility, and share their supplies of the PPE [33].

Skills and competences are required to improve market intelligence and category strategies [31]. During the COVID-19 pandemic, specific skills and competences are required to handle the next disruption [34]. Therefore, it is needed to research which skills and competences are required to implement the strategic changes.

### 4.3. Competences and Skills Used to Implement Strategic Changes

In the previous section, possible strategic changes have been introduced to improve the resilience of supply chains. It is required that PSM professionals have the right skills and competences to make sure these changes can be applied in practice. Currently, PSM professionals focus on objectives that decrease costs, acquire innovations, or improve sustainability [11]. However, PSM managers must shift their focus to implement the strategic changes to improve resilience successfully.

As with many other disciplines, PSM is constantly evolving. Industry 4.0 has shown that the required skills and competences also change over time, especially since PSM has become one of the most crucial departments of the company. Additionally, one of the measures is that a shift towards category management is needed [31]. They mention in their report that implementing this measure requires skills and competences [31]. Moreover, resilient organisations are advised to train employees to handle crises [33].

As mentioned, this study uses the dynamic managerial capabilities theory lens. Adner and Helfat [12] have proposed that three factors root the dynamic managerial capabilities: human capital, social capital, and cognition. Human capital refers to "learned skills that require some investment in education, training, or learning more generally" [12] (p. 1020). Social capital is the result of social relationships [12]. Managerial cognition refers to "managerial beliefs and mental models that serve as a basis for decision making" [12] (p. 1020). Adner and Helfat [12] state that the three factors interact, as depicted in Figure 1.

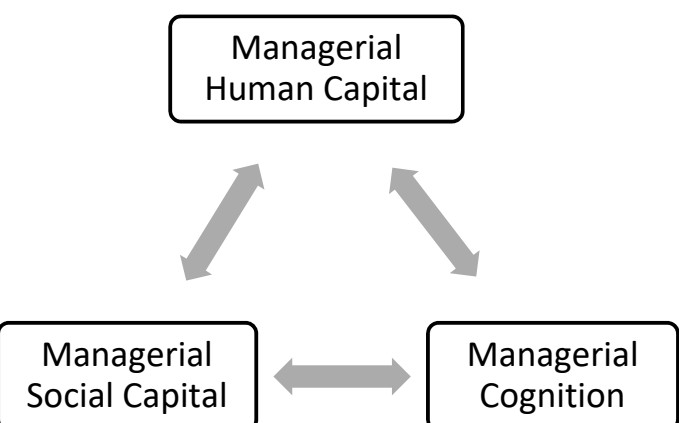

**Figure 1.** Linkages among the roots of managerial capabilities [12] (p. 1022).

Adner and Helfat [12] state that managerial cognition is developed through managerial experience, and that through experience human capital is created. Stek and Schiele [11] studied the necessary and sufficient skills for PSM professionals and distinguished between hard skills (cognition and professional skills) and soft skills (interpersonal skills and intrapersonal traits). They found that soft skills are necessary conditions (i.e., conditio sine qua non) to carry out hard skills [11]. Soft skills (or social capital) are required to develop both hard skills and cognition; the absence of soft skills is a predictor of non-effectiveness in carrying out hard skills.

Stek and Schiele [11] developed a PSM competences taxonomy with 15 factors to solidify the 85 underlying skills and competences. Table 2 shows the factors (grouped competences) [11], which contain a hierarchy. The first four factors (networking, result orientation, imagination, and sellers) and factor 13 (organisational insight and governance) are necessary conditions to carry out the other ten factors' activities [11]. The factors and corresponding items in Table 2 will be applied to determine the skills and competences that PSM professionals need to implement strategic changes to increase resilience.

**Table 2.** PSM skills taxonomy [11] (p. 9).

| Factors | Items |
| --- | --- |
| (1) Networking | Capacity to be empathetic; social manners; loyalty; conscientiousness; honesty; comprehension of complexity; building relations/networking; ability to solve problems; customer-orientation |
| (2) Result orientation | self-assurance; poise; proactivity; result-orientated action-taking; willingness to take risks; capacity to advice; ability to resolve conflicts; power of persuasion |
| (3) Imagination | Creativity; inventiveness; willingness to learn; holistic thinking |
| (4) Sellership | Sellership skills; personality characteristics development (e.g., entrepreneurial); cross-cultural awareness; customer orientation |
| (5) Cross-functional cooperation | Cooperating with the departments such as marketing management; logistics and storage; research and development; production/operations; quality management |
| (6) Forecasting skills | Forecasting of the demand; enterprise resource planning; supply chain analysis |
| (7) Cost focus | Cost reduction techniques; solicit offers; global sourcing; making cost analyses; negotiation; purchasing knowledge |
| (8) Contracting skills | Developing specifications for supplies; contract development (design of contracts); contract management; claims management; evaluate offers and supplier selection; CSR; working together with the legal department |
| (9) Supplier relationship | Supplier relationship management; supply risk management; supplier evaluation; supplier development; early supplier involvement; strategic business partnership; sustainability |
| (10) Innovation sourcing | Innovation sourcing; innovation implementation; category strategy development; stakeholder relationship management; pooling planning and demand; supply market analysis |
| (11) Analytics | Set key performance indicators (KPIs); performance measurement and follow-up; statistical analyses; big data analyses; portfolio analysis support |
| (12) Leadership and personnel management | Purchasing roles and job profiles; personnel selection process; employee integration and development plan; employee performance measurement; leadership/managing personnel; training personnel; managing change processes; working together with the department human resources management |
| (13) Organisational insight and governance | (Understanding how to) add value to the organisation; understanding corporate governance; understanding the position of purchasing in organisation; project management skills; team ability skills; optimisation of purchasing processes; process management |
| (14) Automation | Automation; procurement IT systems/e-procurement applications |
| (15) Technical skills | Technical knowledge of products and production systems; technology planning (knowledge on its own company's technological requirements); commodity and domain-specific knowledge. |

## 5. Discussion

The previous sections have shown the importance of strategic changes in PSM to prepare for future disruptions. Table 1 summarises which strategic changes are required to overcome the actual impacts. To implement the strategic changes, future PSM professionals require new and relevant competences. Stek and Schiele [11] defined the PSM competences shown in Table 2 in their taxonomy of 15 factors and provided quantitative evidence of a competence hierarchy, i.e., that the factors 1–4 and 13, which mainly consist of interpersonal skills and intrapersonal traits, are necessary to carry out competences in other factors. A summarised overview of the competences linked to the strategic changes is provided with this information.

Moreover, Stek and Schiele [11] provided evidence that (9) supplier relationship management and (10) innovation sources are sufficient (independent) conditions leading to effectiveness for different (dependent) PSM targets, namely (A) innovation sourcing and innovation implementation, (B) supplier satisfaction, (C) competitive advantage, and (D) sustainable CSR sourcing. Furthermore, (9,10), and (13) organisational insight and governance are also necessary conditions for (A–E) cost reduction and (F) quality assurance [11]. As Stek and Schiele [11] proposed, the interdependency of competences aligns with the dynamic managerial capabilities theory of Adner and Helfat [12], as shown in Figure 1.

Adner and Helfat [12] state that managerial cognition is developed through managerial experience, and that through experience human capital is created.

In Table 3, a framework is proposed for which competence factors should be allocated to which strategic changes. This framework gives insight into which competences need to be acquired to improve the resilience of a specific change. Since this framework is specified to improve resilience, some competences are more relevant than others.

**Table 3.** Proposed required competences for each strategic change.

| Strategic Change | Competence (Factors) | Explanation |
|---|---|---|
| Diversification/Dual sourcing | (8) Contracting skills (9) Supplier relationship management (10) Innovation sourcing | Supplier relationship management and supply market analysis must evaluate and improve suppliers' diversification. Contracting skills and innovation sourcing are required to define new suppliers' specifications. |
| Vertical integration of supply chains | (9) Supplier relationship management (10) Innovation sourcing (13) Organisational insight and governance | Vertical integration requires an adequate understanding of the organisation's position/insight. Integration of suppliers changes the supplier base, which requires supplier relationship management and supply market analysis. Furthermore, vertical integration requires acquiring factories or suppliers, which requires innovative sourcing since new products must be sourced. |
| Decentralisation of Manufacturing Capacity | (9) Supplier relationship management | Decentralising capacity requires searching and managing multiple suppliers. A good understanding of the capabilities of these suppliers can enhance risk management. |
| Emphasis on Supply Chain Visibility | (14) Automation (15) Technical skills | Information sharing provides good visibility within the supply chain. Developing such systems require technical- and automation skills where multiple levels within the supply chain need to collaborate. |
| Localising Supply Chains | (9) Supplier relationship management | Organised supplier relationship management can provide transparency towards local suppliers having the required capabilities to produce and develop goods. |
| Merging of B2B and B2C/Flexibility in Supply Chains | (4) Sellership (13) Organisational insight and governance | Understanding the organisation's position and capabilities are required for deciding if/how to merge B2B and B2C. Sellership is required for understanding the demands of the current and potential customers in different markets. |
| Investing in Online Distribution Channels | (4) Sellership (10) Innovation sourcing | Exploring possibilities for and creating online distribution channels require innovative solutions in collaboration with customers. Sellership helps to understand the customer's demands. |
| Digital Transformation of Supply Networks | (14) Automation (15) Technical skills | Together with creating visibility, digital transformation of supply chains require automation and technical skills. Organisations required to develop systems for information processing and sharing. |
| Government Policies/Assistance | External measure. No skill necessary | Governmental policies are hardly influenceable |

The connection between strategic changes and competences is based on the match of Table 2 factor items with the conditions of a strategic change. For example, the supplier relationship management competence includes supply risk management related to diversification and dual sourcing. In conclusion, the strategic changes discussed in Section 4.2 and the PSM competence factors are defined in Table 2 [11]. Both are combined in Table 3, showing the strategic changes with the required competences to implement the changes.

As Table 3 shows, most strategic changes can be combined with several factors. Factors (9) supplier relationship management and (10) innovation sources are crucial, since these appear in many strategic changes. The competence factor (13) organisation insight and governance is involved in changing the organisation's position. Competences (14) automation and (15) technical skills are essential for the technical and digital strategic changes.

## 6. Conclusions

In this literature study, the impacts of COVID-19 on supply chain resilience are researched and combined with the PSM competence literature. The following question was central in this study: *"Which strategies and corresponding competences are needed to improve supply chain resilience?"*

Resilience can be defined in many ways, but it is generally the ability to absorb a disruption. In this case, the disruption is COVID-19 in supply chains. The Kraljic matrix illustrates that supplies with initially slight supply risks shift "to the right" and develop significant supply risks during such disruption. Current trends such as globalisation and risk management that did not contribute to more resilient supply chains, meaning supply chains have become more vulnerable to disturbances. The main optimisation factor of a supply chain was previously (cost) efficiency.

However, during the COVID-19 pandemic, supply and demand shocks occurred that could not be handled. Supply shortages arose due to the significantly increased demand for certain products and scarce resources. This led to a disrupted state of the global production and logistics sector.

To reduce the impacts of the next disturbance, companies should expand and diversify their supplier database and localise their supply chains. Moreover, organisations should emphasise transparency and visibility within the entire supply chain and start a digital transformation of supply networks. Furthermore, companies should look for more flexibility in their supply chains to respond to changing demands. Implementing these measures will reduce risks in the supply chain, which will increase resilience. Contrary to the trend over the past few decades whereby supply chains are optimised based on (cost) efficiency, the trend should change towards giving resilience a higher weight in the optimisational strategies of supply chains.

Specific skills and competences are needed to implement these strategic changes. The essential skills and competences are (9) supplier relationship management, (10) innovation sourcing, (14) automation, and (15) technical skills. Moreover, (1) networking, (2), result orientation, (3) imagination, (4) sellership, and (13) organisational insight and governance are necessary conditions to carry out the competences, as proposed in Table 4. PSM should invest in training their staff in these skills and competences to improve resilience within their supply chains. In Table 3, a framework is proposed for the skills and competences needed for the proposed strategic changes. Table 4 shows the primary connection between the four different impacts of COVID-19, the strategic changes, and the corresponding competences needed. All other strategic changes and competences needed help to create more resilient supply chains are not directly coupled to one of the four impacts of COVID-19.

**Table 4.** Connecting COVID-19 impacts with strategic changes and essential competences.

| Impact of COVID-19 | Strategic Change | Competence (Factors) Needed |
|---|---|---|
| Supply shocks | Diversification/dual sourcing, Localising supply chains | (8) Contracting skills, (9) Supplier relationship management, and (10) Innovation sourcing |
| Demand shocks, Bullwhip effect | Emphasis on supply chain visibility, Increased resilience through stockpiles | (14) Automation and (15) Technical skills |
| Transportation and requirement costs | Localising supply chains | (9) Supplier relationship management |

## 7. Future Research and Managerial Implications

According to the presented framework, top managers should consider training specific skills and competences for PSM professionals to improve resilience within supply chains. PSM professionals should be equipped with these skills and competences to cope with future disruptions. The recommendation is to train the PSM professionals within companies and adapt curricula in higher education by integrating the training of skills and competences needed for a more resilient future. The literature has provided evidence that transferable, non-cognitive, or soft skills can be developed in higher education [35–37].

A taxonomy of PSM competences was defined by Stek and Schiele [11]. They differentiate between sufficient and necessary competences. Interestingly, they found that carrying out (9) supplier relationship management, (10) innovation sourcing, (14) automation, and (15) technical skills requires multiple other competences, which were grouped in five fac-

tors, namely (1) networking skills, (2) result orientation, (3) imagination, (4) sellership skills, and (13) organisational insight and governance [11]. However, there is evidence that PSM job requirements are subject to cultural influence; in other cultural areas, nations, or regions, other requirements, mainly transferable, non-cognitive, or soft skills, are demanded [38].

These five competency factors are typically associated with interpersonal human-to-human skills and intrapersonal character traits or transferable, non-cognitive, or soft skills. The literature shows that PSM bachelors and masters courses do not, with limited exceptions, formalise these interpersonal skills and intrapersonal learning objectives [39,40]. Indeed, Fawcett and Rutner [41] provided evidence that higher education is "not evolving at the pace and in the way expected by professionals" [41] (p. 181).

The COVID-19 pandemic makes it even more evident that developing supplier risk management, interpersonal skills, and intrapersonal traits in higher education (and in-company training) needs more attention. A suggestion for further research is assessing what effect a focus on resilient supply chains instead of cost-efficient supply chains has on higher educational programmes. There is evidence that transferable, non-cognitive, or soft skills are necessary (but not sufficient) to carry out professional or hard skills, meaning that an absence of the first leads to a problematic execution of PSM activities [11].

**Author Contributions:** Conceptualization, K.S.; methodology, all authors; writing—original draft preparation, J.K., J.S. and M.Z.; writing—review and editing, all authors; supervision, K.S. All authors have read and agreed to the published version of the manuscript.

**Funding:** This research received no external funding.

**Institutional Review Board Statement:** Not applicable.

**Informed Consent Statement:** Not applicable.

**Conflicts of Interest:** The authors declare no conflict of interest.

**Appendix A**

**Table A1.** The literature on COVID-19 supply chain issues.

| Author | Title | Content and Unique Contribution |
|---|---|---|
| Chowdhury, Paul [4] | COVID-19-pandemic-related supply chain studies: A systematic review | The article conducts a systematic review of the literature available (on or before 28 September 2020) on the COVID-19 pandemic for several supply chain disciplines. The unique contribution is that the article combines the literature on COVID-19 with the prior outbreak literature to provide research gaps and future research questions. |
| Finkenstadt and Handfield [32] | Blurry vision: Supply chain visibility for personal protective equipment during COVID-19 | The article explores supply chain visibility challenges in the context of COVID-19 and offers insights, models, and potential solutions to increase supply chain visibility. The unique contribution is that the article provides a framework for enhancing PPE visibility that can easily be generalised. They also generally enhance uncovering hidden stocks in the supply chain spectrum. |
| Golan, Jernegan [15] | Trends and applications of resilience analytics in supply chain modelling: systematic literature review in the context of the COVID-19 pandemic | The article reviews the supply chain resilience literature that focuses on resilience modelling and quantification and explains why a comprehensive approach to network resilience quantification encompassing the supply chain in the context of other social and physical networks is needed to address the emerging challenges in the field of supply chain resilience. The unique contribution is that the article elaborates on supply chain resilience quantification trends instead of general supply chain resilience. |

**Table A1.** *Cont.*

| Author | Title | Content and Unique Contribution |
| --- | --- | --- |
| Handfield, Finkenstadt [33] | How Business Leaders Can Prepare for the Next Health Crisis | The article provides strategies that can reduce organisations' vulnerability across various threats. The unique contribution is that the article focuses on critical emergency resources to keep people and communities safe in a crisis and the call for organisations to strive for immunity. |
| Handfield, Finkenstadt [34] | A Commons for a Supply Chain in the Post-COVID-19 Era: The Case for a Reformed Strategic National Stockpile | The article reflects on the current response deficiencies and offers a model for a national contingency supply chain cell (NCSCC) to manage the medical materials supply chain in emergencies. The unique contribution is that the article does not limit the response deficiencies to one specific purchase category, region, or border, but it takes a broader view. |
| Harland, Knight [42] | Practitioners' Learning about Healthcare Supply Chain Management in the COVID-19 Pandemic: A public procurement perspective | The article captures learning from practitioners in "real-time" to frame and inform capacity building across healthcare systems with varying PSM maturity. The unique contribution is that the article uses the awareness–motivation–capability framework to provide a comprehensive overview of healthcare procurement from a system perspective. |
| Ivanov [28] | Predicting the impacts of epidemic outbreaks on global supply chains: A simulation-based analysis on the coronavirus outbreak (COVID-19/SARS-CoV-2) case | The article analysis observed and predicted short- and long-term impacts of epidemic outbreaks on the supply chains and managerial insights. It helps identify successful and wrong elements of risk mitigation and recovery policies. The unique contribution is that the author conducts a simulation study on COVID-19 to provide insights into the elements of risk mitigation and recovery policies. |
| Magableh [24] | Supply Chains and the COVID-19 Pandemic: A Comprehensive Framework | The article examines the impact of COVID-19 on supply chains regarding its disruptions, associated challenges, and trends. It provides a framework for the supply chain towards a future global value chain and continuous improvements. The unique contribution is that the study identifies, categorises, and frames the essential factors and their relationship in one framework that can be generalised to other industries. |
| Mena, Karatzas [16] | International trade resilience and the COVID-19 pandemic | The article investigates country-level trade resilience during the first wave of the COVID-19 pandemic and identifies factors that strengthen or weaken international trade resilience. The unique contribution is that it analyses the impact of COVID-19 on international trade resilience. |
| Pujawan and Bah [43] | Supply Chains under COVID-19 Disruptions: Literature Review and Research Agenda | The article presents a literature review that addresses supply chain disruptions due to COVID-19 and discusses the significant findings or recommendations. The unique distribution is that the article provides an overview of the changes in disruption mitigation strategies in different areas of supply chain management. |
| Rio-Chanona, Mealy [23] | Supply and demand shocks in the COVID-19 pandemic: an industry and occupation perspective | The article provides quantitative predictions of first-order supply and demand shocks for the US economy associated with the COVID-19 pandemic at the level of individual occupation and industries. The unique contribution of the article is that it solely focuses on two specific impacts of COVID-19, supply and demand shocks. |

**Table A1.** *Cont.*

| Author | Title | Content and Unique Contribution |
| --- | --- | --- |
| Sajjad [5] | The COVID-19 pandemic, social sustainability, and global supply chain resilience: A review | The article examines the impacts of COVID-19 on global supply chain sustainability. The unique contribution is that the article focuses on creating more resilient and sustainable supply chains in the post-COVID world. |
| Sarkis [44] | Supply chain sustainability: learning from the COVID-19 pandemic | The article provides research guidance for investigating sustainability in supply chains in a post-COVID-19 environment. The unique contribution is that the article focuses solely on sustainability. |
| Shi, Liu [6] | Present and future trends of supply chain management in the presence of COVID-19: a structured literature review | The article shows a structured literature review on supply chain-related issues and provides a research agenda considering supply chain disruptions and 3Rs in the supply chain. The unique contribution is that the article provides new trends for the theoretical and applied research in disaster management, risk management, and incident management. |
| van Hoek [1] | Research opportunities for a more resilient post-COVID-19 supply chain: closing the gap between research findings and industry practice | The article suggests a pathway for closing the gap between supply chain resilience research and efforts in the industry to develop a more resilient supply chain. The unique contribution is that the article helps close the gap between supply chain resilience research and efforts in the industry to improve supply chain resilience. |
| Spieske and Birkel [3] | Improving supply chain resilience through industry 4.0: A systematic literature review under the impressions of the COVID-19 pandemic | The article provides a review of the implementation of Industry 4.0. to increase supply chain resilience based on the COVID-19 pandemic. The unique contribution is that the article provides findings that can be used to guide Industry 4.0. investment decisions. |
| Zhu, Chou [22] | Lessons Learned from the COVID-19 Pandemic Exposing the Shortcomings of Current Supply Chain Operations: A Long-Term Prescriptive offering | The article addresses the relationship between supply chain operations and the COVID-19 pandemic and provides recommendations to mitigate current consequences and improve the resilience needed to weather similar potential shortages in the future. The unique contribution is that the article focuses on the long-term prescriptive offering instead of the short-term offering. |

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
