# Peer review of "Which Strategies and Corresponding Competences Are Needed to Improve Supply Chain Resilience: A COVID-19 Based Review"

_logistics, 2017_

Round 1
Reviewer 1 Report
Dear authors, thank you for the opportunity to read and review your manuscript. I do think the topic is very timely and you present interesting points. However, I have couple of major points that I would like see improved in the manuscript, and couple of minor suggestions/errors that need fixing.
First of all, I think your manuscript needs a separate section on methodology. You say in the abstract that a literature review is your method, but we don't get to see how the method was approached. You even claim that your review is systematic at one point. I assume you applied some method and structure to the process of identifying, reading and structuring the papers that you reviewed. Thus, I would like to see what keywords, databases, journals, dates, inclusion/exclusion criteria, etc. were utilized in the review. For example, what you do in the paper seems quite similar to the Theory, Methodology, and Context (TMC) framework (Paul et al., 2017), which you could try to adapt.
I also think (based on Table A5) that the review was focused only on the impacts of COVID, not RQs 1 and 2 (that would be a completely different paper with completely different sources). Thus, I would suggest removing the RQs since they're not the point of your paper and the phenomenon of SC resilience can instead be introduced in a separate theoretical section.
I actually quite like the results in sections 2.3 and 2.4. However, I would suggest not just listing examples from the literature, but also providing some additional points how that particular strategic change would mitigate some issue raised by COVID (e.g., the subsection on Decentralisation of Manufacturing Capacity) is very short.
Again, subsection 2.5 is quite nice and informative. However, from methodological perspective, we're lacking information about how did you came up with Table 3. Did this somehow came out of your literature review, or is it authors' own framework? How were the competences and changes combined? I find the results to be a bit random, perhaps due to very short explanation.
I find the ordering of sections 3 and 4 confusing, I must admit I have never seen discussion after conclusion. Should not, by definition, conclusion section conclude the paper? Anyway, both of those sections are very short and not that informative. Perhaps they could even be combined into one "Discussion and conclusion" section. The discussion section is particularly shallow and focuses just on a fraction of the findings (the skills/competences).
Minor points/suggestions:
- in section 2, the subheadings are not very consistent and range from short (2.1) to long (2.4) to questions (2.5). I would suggest unifying the headings.
- on line 229, you say "To reduce the reliability of China", which does not make too much sense. Perhaps you've meant "To reduce the reliance on China"?
- I don't think it is ethically correct to screenshot the whole figure from the website in Figure 1. I would suggest going back to the original paper, and using your own interpretation as a basis for the figure.
- overall, I think the language needs to be improved to be more precise/rigorous. E.g., what are "last few decades" in RQ2? 2? 3? 5? Could be fixed by providing the time frame of the literature review.
Author Response
Dear reviewers and editors,
Thank you for allowing us to submit a revised draft of our manuscript titled “The impact of COVID-19 on supply chain resilience: improving supply chains for a more resilient future” to Logistics. We appreciate the time and effort you dedicated to providing valuable feedback on our manuscript. We have been able to incorporate changes to reflect most of the suggestions provided by the reviewers.
Below is a point-by-point response to the reviewers’ comments and concerns. In addition to the above comments, spelling and grammatical errors pointed by the reviewers have been corrected.
We look forward to hearing from you in due time regarding our submission and to respond to any further questions and comments you may have.
Sincerely,
The authors
Comments from reviewer 1
Comment 1: “First of all, I think your manuscript needs a separate section on methodology. You say in the abstract that a literature review is your method, but we don’t get to see how the method was approached. You even claim that your review is systematic at one point. I assume you applied some method and structure to the process of identifying, reading and structuring the papers that you reviewed. Thus, I would like to see what keywords, databases, journals, dates, inclusion/exclusion criteria, etc. were utilized in the review. For example, what you do in the paper seems quite similar to the Theory, Methodology, and Context (TMC) framework (Paul et al., 2017), which you could try to adapt.”
Response 1: We agree with this comment and added a methodology section (section 2).
Comment 2: “I also think (based on Table A5) that the review was focused only on the impacts of COVID, not RQs 1 and 2 (that would be a completely different paper with completely different sources). Thus, I would suggest removing the RQs since they’re not the point of your paper and the phenomenon of SC resilience can instead be introduced in a separate theoretical section.”
Response 2: We agree with this comment. We put research questions 1 and 2 in a theoretical background section (section 3) since this is not the main point of our paper but import knowledge to know in advance.
Comment 3: “I actually quite like the results in sections 2.3 and 2.4. However, I would suggest not just listing examples from the literature, but also providing some additional points how that particular strategic change would mitigate some issue raised by COVID (e.g., the subsection on Decentralisation of Manufacturing Capacity) is very short.”
Response 3: We agree with your comment. However, our study does not focus on how strategic changes can increase resilience. The proposed strategic changes are a result of the literature review. Our main result is our proposed framework in which we propose the skills and competences needed to implement these strategic changes.
Comment 4: “Again, subsection 2.5 is quite nice and informative. However, from methodological perspective, we’re lacking information about how did you came up with Table 3. Did this somehow came out of your literature review, or is it authors’ own framework? How were the competences and changes combined? I find the results to be a bit random, perhaps due to very short explanation.”
Response 4: We agree with your comment. We have changed the position of Table 3 from the literature review chapter towards the discussion and clearly described that this is indeed the authors’ framework. We also added an additional explanation about how we came up with the proposed framework.
Comment 5: “I find the ordering of sections 3 and 4 confusing, I must admit I have never seen discussion after conclusion. Should not, by definition, conclusion section conclude the paper? Anyway, both of those sections are very short and not that informative. Perhaps they could even be combined into one “Discussion and conclusion” section. The discussion section is particularly shallow and focuses just on a fraction of the findings (the skills/competences).”
Response 5: We have changed the ordering of sections 3 and 4. We now start with the discussion, in which we propose a new framework regarding the needed skills and competences to implement the proposed strategic changes into supply chains. Consequently, we conclude our paper by answering the research questions.
Comment 6: “in section 2, the subheadings are not very consistent and range from short (2.1) to long (2.4) to questions (2.5). I would suggest unifying the headings.”
Response 6: We agree with this comment. The heading of the main sections is made short, whereas the heading of the subsections contains more specific content.
Comment 7: “I don’t think it is ethically correct to screenshot the whole figure from the website in Figure 1. I would suggest going back to the original paper, and using your own interpretation as a basis for the figure.”
Response 7: We agree with the comment and have used the original paper to construct the Kraljic Matrix and visualise the shift towards higher supply risk.
Comment 8: “Overall, I think the language needs to be improved to be more precise/rigorous. E.g., what are “last few decades” in RQ2? 2? 3? 5? Could be fixed by providing the time frame of the literature review.”
Response 8: We have changed the unclear terms in the text. We agree with the comment.
Comments from reviewer 2
Comment 1: “The motivation for this study in the Introduction is not strong enough. Throughout the introduction, authors briefly discuss the importance of supply chain resilience and COVID-19 based disruptions. My chief concern is that the originality of the paper is not clear. The introduction section is confusing; the following structure is recommended: (i) establish the importance of research, (ii) establish a theory-based gap, (iii) research question, (iv) contribution, and (v) paper structure.”
Response 1: We agree with this comment and therefore changed the introduction to the proposed structure. We also emphasise the originality of the paper and the theory-based gap. The theory we now use as a lens is the Dynamic Managerial Capabilities theory, as our main result is a framework in which the needed skills and competences are described. It directly links to the dynamic managerial capabilities theory.
Comment 2: “There is no research nor theoretical background about the proposed model. This paper does not change, challenge or fundamentally advance our knowledge of the concepts, relationships, models or theories embedded in the relevant literatures of supply chain management, resilience management, and strategic management. The purpose of research/theoretical background is to create thought and dialogue surrounding a business process management phenomenon in a way that would not normally be anticipated from extrapolations of existing work, thereby advancing future work in an important and useful way.”
Response 2: We agree with this comment. We put research questions 1 and 2 in a theoretical background section (section 3) since this is not the main point of our paper but import knowledge to know in advance. Also, a theory lens is used to create thought and dialogue about improving supply chain resilience in terms of strategic changes and corresponding competencies and skills needed according to lessons learned from COVID-19. The theory lens used is the dynamic managerial capabilities theory lens. The theory lens is explained in the introduction and used in the remainder of the article.
Comment 3: “While the authors have stated: “Two significant gaps in the field of supply chain resilience are (1) no outline on which strategy to deal with, so which disruptions solve which impact on the disruption and (2) there are no studies that explore the challenges and requirements associated with implementing resilience strategies.” on Pg. 2 Line 48-51, the proposed investigation approaches and/or results do not resolve the raised issue. Authors are highly recommended to do their diligence in building a theoretical (research) background to continue their journey on improving this manuscript.”
Response 3: The authors see the value of the comment. We have changed our description of the gap our study fills, adding a theoretical background section.
Comment 4: “Authors are recommended to refer to the following articles regarding systematic literature review and supply chain resilience topic:”
Response 4: We have used an article by Snyder (2019) to construct our methodology. This article focuses on Literature review as a research methodology and gives an overview and guidelines.
Comment 5: “The research & managerial implications sections are missing. The main purpose of any research is to contribute to Knowledge and lessons for academicians/practitioners, and to discuss future research avenues by discussing limitation which require more improvement.”
Response 5: The authors agree with this comment. As a result, we have added a Future research & Managerial Implications section, in which we describe the main consequences of our proposed framework for practice and give suggestions for future research.
Comment 6: “It is recommended a professional copy-editing before re-submission.”
Response 6: a professional copy-editing has been performed before re-submission.
Reviewer 2 Report
Please refer to the attached file for comments.

Author Response

(The authors gave the same response as above.)

Round 2
Reviewer 1 Report
Dear authors, I appreciate the effort that you've taken in addressing mine and other reviewer's comments. I believe the manuscript is almost ready to be published, and have only one small suggestion for your (new) methodology section:
- could you sum up the number of papers that you've included/excluded in each step of the review? E.g. after search, how many were excluded after screening abstracts, how many were added from other sources (what were those sources?) See for example the recent paper by Mitrega et al. (2022) for how this information can be provided visually.
- related to that, the inclusion/exclusion criteria are very blurry. Could you perhaps give a few examples of papers that were found using your criteria, but were subsequently removed from the database, with reasons for the removal?
Author Response
Dear reviewer,
Thank you for your appreciation and immediate review. We have now included the steps taken in the literature review process. Moreover, a new proofread solved minor issues in the text.
Reviewer 2 Report
This reviewer appreciates the efforts made by the authors, and believes that the readers of Logistics can benefit from the outcome of this research.
Author Response
Dear reviewer,
Thank you for your appreciation and your immediate review. A new proofread solved minor issues in the text.